# In Silico Analysis of a *Drosophila* Parasitoid Venom Peptide Reveals Prevalence of the Cation–Polar–Cation Clip Motif in Knottin Proteins

**DOI:** 10.3390/pathogens12010143

**Published:** 2023-01-14

**Authors:** Joseph Arguelles, Jenny Lee, Lady V. Cardenas, Shubha Govind, Shaneen Singh

**Affiliations:** 1Department of Biology, Brooklyn College, Brooklyn, NY 11210, USA; 2Department of Biology, The City College of New York, New York, NY 10031, USA; 3PhD Program in Biochemistry, The Graduate Center of the City University of New York, New York, NY 10016, USA; 4PhD Program in Biology, The Graduate Center of the City University of New York, New York, NY 10016, USA

**Keywords:** host–parasite, *Drosophila*, *Leptopilina*, cysteine knot, knottin fold, Cation–Polar–Cation clip, antimicrobial peptide, heparin-binding motif, miniproteins

## Abstract

As generalist parasitoid wasps, *Leptopilina heterotoma* are highly successful on many species of fruit flies of the genus *Drosophila*. The parasitoids produce specialized multi-strategy extracellular vesicle (EV)-like structures in their venom. Proteomic analysis identified several immunity-associated proteins, including the knottin peptide, LhKNOT, containing the structurally conserved inhibitor cysteine knot (ICK) fold, which is present in proteins from diverse taxa. Our structural and docking analysis of LhKNOT’s 36-residue core knottin fold revealed that in addition to the knottin motif itself, it also possesses a Cation–Polar–Cation (CPC) clip. The CPC clip motif is thought to facilitate antimicrobial activity in heparin-binding proteins. Surprisingly, a majority of ICKs tested also possess the CPC clip motif, including 75 bona fide plant and arthropod knottin proteins that share high sequence and/or structural similarity with LhKNOT. Like LhKNOT and these other 75 knottin proteins, even the *Drosophila* Drosomycin antifungal peptide, a canonical target gene of the fly’s Toll-NF-kappa B immune pathway, contains this CPC clip motif. Together, our results suggest a possible defensive function for the parasitoid LhKNOT. The prevalence of the CPC clip motif, intrinsic to the cysteine knot within the knottin proteins examined here, suggests that the resultant 3D topology is important for their biochemical functions. The CPC clip is likely a highly conserved structural motif found in many diverse proteins with reported heparin binding capacity, including amyloid proteins. Knottins are targets for therapeutic drug development, and insights into their structure–function relationships will advance novel drug design.

## 1. Introduction

In host–parasite interactions, the parasite must provide offensive pressure while simultaneously keeping the host alive to sustain viable offspring [1,2]. Such conflicting goals have led to the evolution of virulence factors with diverse molecular strategies while maintaining high host specificity. Parasitic Hymenoptera make up more than a million species, are ubiquitous on the planet, and parasitize on a variety of arthropods, mainly insects [3]. Wasps attacking insect hosts have developed an assortment of behavioral and biochemical strategies for success and remain under constant co-evolutionary pressure with their hosts [4,5,6,7].

The *Leptopilina*/*Drosophila* system is an emerging model for understanding the molecular foundations of anti-wasp responses and wasp virulence strategies [8]. To suppress host immunity, *Leptopilina* females introduce venom factors into their larval hosts during oviposition [4,9,10]. The well-defined innate immune mechanisms in *Drosophila* [11,12,13] provide a valuable context to study the effects of these wasp venom factors. Infection by the specialist *L. boulardi* activates the humoral and cellular immune arms controlled by the fly’s conserved Toll-NF-κB signaling pathway. The expression of antimicrobial peptide (AMP) genes such as *drosomycin* is activated after *L. boulardi* infection [14]. Toll-NF-κB signaling also controls blood cell division and development [15,16], steps that control encapsulation and death of wasp eggs [17,18]. In contrast, infection by the generalist *L. heterotoma* suppresses both immune pathways [9,14], and almost all larval blood cells are destroyed by factors in the wasp venom [19,20].

In this study, we analyze the theoretical structure of an *L. heterotoma* peptide, called LhKNOT, identified within organelle-like secretions in the *L. heterotoma* venom but not in *L. boulardi* venom [21]. These immune-suppressive organelles include more than 350 proteins and possess a proteomic profile of eukaryotic microvesicles and a plethora of immunity-related proteins [21,22,23]. LhKNOT adopts the “Inhibitor Cysteine Knot” (knottin) motif [24]. The knottin motif is broadly conserved in a range of taxa, from plants [25,26] to snails [27]. These small peptides (<60 residues) are functionally diverse. Knottins are used offensively, in spider toxins [28] and cone snail venom [27], as well as defensively, e.g., in wound healing activities of cacti [26], and antimicrobial host defense in plants [25,26,29] and invertebrates [30].

The broad range of knottin activity is attributed to the overall structure of its fold, which relies solely on a conserved motif of six cysteine residues that takes advantage of these cysteines’ unique ability to form disulfide bridges. The knottin motif occurs when cysteine residues are observed in a C1-C2-C3-C4-C5-C6 pattern (where “-“ represents a number of highly variable residues). Disulfide bridges are then formed in a similarly conserved fashion (C1-C4, C2-C5, and C3-C6), resulting in C1-C4 and C2-C5 forming a macrocycle, while C3-C6 occurs within this macrocycle, resulting in a “knotted” topology. As this knotted topology largely hinges on the conservation of the six cysteine residues, the knottin motif allows for significant variation in the overall sequence [30]. The combination of their small size, high tolerance of sequence variance, and observed structural stability has made knottins a popular scaffold for drug design [30], potentially even for cancer treatments [31].

Various lines of experimental evidence demonstrate or suggest antimicrobial capabilities for different knottins [25,26,29,30,32,33] or gated ion channel interactions [28,34], although the molecular mechanisms by which these functions are carried out are not well understood. Furthermore, it is not known if these antimicrobial functions share a common mechanism of action. Given its discovery from immune-suppressive particles from fly parasites, our goals were to (a) identify LhKNOT-related sequences from diverse organisms, including *L. heterotoma*, *L. boulardi*, and *Ganaspis* parasitoids, that attack *Drosophila* and (b) identify any novel structural motif(s) present in the known or new LhKNOT homologs with the hope that the prediction of such a conserved motif might suggest possible functions for LhKNOT and guide future experimental research.

We show that a theoretical model of LhKNOT possesses the hallmarks of the knottin fold with three antiparallel beta sheets, constrained by disulfide bridges. The LhKNOT model contains a CPC clip, capable of interacting with heparin, a negatively charged glycosaminoglycan (GAG) of significant medical value. LhKNOT also has the potential for interacting with other GAGs, such as keratan sulfate (a GAG that is highly sulfated, similar to heparin) and hyaluronic acid (the single GAG representative that does not require sulfation) [35]. Furthermore, structural homologs of LhKNOT (i.e., bona fide knottin motif proteins) also possess the CPC clip motif that can dock with heparin. Surprisingly, like LhKNOT and these other bona fide knottin proteins, even the *Drosophila* Drosomycin antifungal peptide shares this CPC clip motif in its structure. This motif was also identified in 41 out of 46 additional LhKNOT homologs from *L. heterotoma*, *L. boulardi*, and *Ganaspis* spp. wasps successfully on *D. melanogaster*. These in silico-based observations suggest that the CPC clip motif, inherent in many knottins, may provide the structural basis for modifying immune or virulence functions. We speculate on a possible antimicrobial or virulence-related function for the wasp LhKNOT. The abundance of the knottin fold and its concurrence with the CPC clip motif in most of the newly identified knottin peptides from three *Drosophila* parasitoids suggests an important yet to be determined role for them in parasitoid physiology and/or in the host–parasite interactions.

## 2. Results

### 2.1. The LhKNOT Sequence Shows Similarity to Insect and Plant Antimicrobial Peptides

A BLASTp search of the nr and TSA databases using the full-length LhKNOT sequence (60 amino acids) showed a limited number of significant hits, primarily annotated as “antimicrobial peptides”, from plants and insects. A few fungal sequences were also identified. All results displayed less than 60% amino acid identity to LhKNOT. The top match for LhKNOT in the nr database was an antimicrobial peptide from the common beetle, *Callosobruchus maculatus*, while the top hit in the TSA database was an antimicrobial peptide from the wheat curl mite *Aceria tosichella*. The conservation was most prominent in the region of the only sequence motif associated with LhKNOT, the cysteine knotted ‘antifungal peptide’ signature (pfam11410 domain, E = 2.8 × 10^−3^). After running PSI-BLAST for detection of remote homologs, additional sequences were revealed. PSI-BLAST showed similar results with common ice plant, *Mesembryanthemum crystallinum*; red mites, *Dinothrombium tinctorium*; parasitoid wasp, *Trichogramma pretiosum*; click beetle, *Ignelater luminosus*; parasitoid wasp, *Trichogramma brassicae*; common pollen beetle, *Brassicogethes aeneus*; ash borer, *Agrilus planipennis*; and sawfly, *Neodiprion lecontei*. Using the Hidden Markov Models-driven HMMER webserver as an alternative method of detecting remote homologs, we identified five fungal hits from *Akanthomyces lecanii*, *Cordyceps javanica*, *Cordyceps confragosa*, *Rosellinia necatrix*, and *Beauveria bassiana*.

A search of the PDB for similar sequences retrieved only two significant results, one for a knottin-type antifungal peptide Alo-3 from the insect *Acrocinus longimanus* and the second from the disulfide-constrained wound healing peptide pB1 from *Pereskia bleo*. The details of all close and remote homologs identified using the different sequence similarity approaches are summarized in Appendix A.

An alignment of LhKNOT to the known structures of typical knottins as well as identified homologs showed the conservation of the six cysteines that form the three disulfide bridges in LhKNOT (Figure 1A). On its N-terminus, LhKNOT contains a long, helical segment predicted to function as a signal sequence; internally, it contains three beta-sheets, as expected of the secondary structure of knottin peptides (Figure 1B).

Additionally, an assessment of sequence similarity of LhKNOT targeted to the available *L. heterotoma* and *L. boulardi* transcripts and from another *Drosophila* parasitoid, *Ganaspis* spp. (see Methods), revealed an abundance of related sequences. For *L. heterotoma* (*Lh14*), we identified 16 additional *Lh14* knottin proteins (Appendix A). For *L. boulardi*, we identified 22 knottin proteins (Appendix A). Six LhKNOT homologs were similarly identified from the *Ganaspis hookeri* (G1) and two LhKNOT homologs were found in the protein predictions made from the *G. brasiliensis* (VA) genome (Appendix A). Figure 2 shows the conservation of these parasitoid knottin sequences with LhKNOT. Thus, at least the few *Drosophila* parasitoids sampled in this study appear to encode multiple knottin proteins. Additional such knottin peptides may be encoded by these wasps, and systematic proteomic/bioinformatic approaches are needed to obtain a comprehensive list. The functions of these putative knottins in parasite physiology remain unknown.

### 2.2. The Three-Dimensional Structure of LhKNOT

The typical structural fold of a knottin peptide is defined by six cysteines involved in three disulfide bonds to form a cysteine-knotted three-stranded antiparallel beta-sheet [25]. An important characteristic of this fold is the connectivity of the cysteine residues to form the disulfide bridges and the disulfide bridge of cysteines 3 and 6 going through disulfides 1–4 and 2–5 to make a special disulfide through disulfide knot (Figure 1A,C). This disulfide, through a disulfide knot feature, distinguishes the knottin fold from other cysteine knot proteins. Other than the conserved cysteines, the rest of the amino acid sequence can be quite variable, and therefore, only a limited number of knottins are identified by sequence similarity approaches for LhKNOT (see the previous section and Appendix A). Fold recognition algorithms and template-based modeling approaches, on the other hand, retrieved several diverse knottins as suitable candidates for modeling LhKNOT. Among the various LhKNOT models generated, the best model was a multi-template-based model of LhKNOT created using the program HHpred [37] with structural restraints derived from the top three templates, Alo-3 (PDB: 1Q3J) [30], omega-agatoxin-IVA (PDB: 1IVA) [28], and PAFP-S (PDB: 1DKC) [25]. Knowledge-based energy profiles calculated using ProSA-web [38] for this selected model showed low energies and a z-score of −4.53, which is comparable to solved structures. Verify3D [39] assessed this multi-template-based model with a passing score (100.00% of the residues in the model have an averaged 3D-1D score greater than or equal to 0.2). The VoroMQA [40] score of 0.325 is likely a reflection of the small size of the peptide and the scores of the templates were in the same range (Appendix A). This multi-template-based model typifies the knottin fold with three antiparallel beta strands, constrained by disulfide bridges and the expected connectivity of disulfide bridges (Figure 1D). KNOTER3D, a tool in the KNOTTIN Database [36] corroborated that this model has a bona fide knottin fold and confirmed the previously identified disulfide bonding pattern in Figure 1D.

### 2.3. LhKNOT Shows the Presence of a CPC Clip, Capable of Interacting with Heparin and Other GAGs

To search for potential LhKNOT functions, structure–function correlations of close structural homologs of LhKNOT were investigated. A knottin protein from the cactus *Pereskia bleo* (PDB: 5XBD) [26] possesses a heparin-binding motif known as the CPC clip. This structural motif forms a geometrically constrained clip of two cationic and one polar residues that interact with heparin by salt bridges and hydrogen bonds clamping it in position within the positively charged surface of the clip [41]. A structural superposition of the cactus knottin and LhKNOT showed that the two structures are highly similar, with an overall difference of less than 2Å. Docking analysis of LhKNOT with heparin, using ClusPro’s specific parameters for heparin as a ligand, revealed that for LhKNOT, the polar interactions fitting the CPC clip motif occur in residues R1-S3-R14 (Figure 3A). Measured distances (both between alpha-carbons and centers of mass of participating residue side chains) differed from those of the CPC clip by less than 1Å, suggesting the presence of a putative CPC clip in LhKNOT (Figure 3B). The surface electrostatics of the modeled LhKNOT also showed the presence of the expected cationic binding pocket formed by the CPC clip [41,42] (Figure 3C).

To assess the CPC clip’s ability to interact with other GAGs, additional docking analysis was performed with LhKNOT and keratan sulfate (PDB ID: 1KES) and with LhKNOT and hyaluronic acid (PDB ID: 1HYA). In both cases, the positively charged binding surface created by the CPC clip bound to the negatively charged molecules. Ligplots generated by PDBSum confirmed an association of the GAGs with the CPC-forming residues (Appendix A).

### 2.4. Other Structural Homologs of LhKNOT Also Contain the CPC Clip Motif

Presence of the CPC clip in both LhKNOT and its structural homolog from *Pereskia bleo* (pB1) suggested the possibility of a broader conservation of the CPC clip within the knottin family. An analysis of 21 close structural homologs of LhKNOT with experimentally solved PDBs in the KNOTTIN database revealed that, in every case, the peptides possess the CPC clip motif. For most peptides (e.g., *T. tridentatus*), the motif parameters deviated less than 1Å from the core knottin fold, while for others (e.g., the offensive knottins), the deviation was up to a margin of 3Å (Appendix A). Similarly, modeled knottin proteins of the identified homologs showed that almost all of them possessed the CPC clip.

Sixteen additional sequences from *L. heterotoma*, 22 sequences from *L. boulardi*, and 8 sequences from *Ganaspis* spp., with no known 3D structures, were modeled and analyzed for the presence of the CPC clip motif. While 13 of the 16 putative *L. heterotoma* knottins and 20 of 22 putative *L. boulardi* knottins revealed a CPC clip motif, all *Ganaspis* sequences contained the CPC clip (Appendix A). Interestingly, the seven homologs that did not exhibit a CPC clip motif belong to (a) the common pollen beetle sequence (*B. aeneus* (CAH0563829.1), (b) fungal sequence *R. necatrix* (A0A1S8A723), and (c) three *L. heterotoma* knottins 4,5,6 and *L. boulardi* knottins 3 and 20 (Appendix A).

A survey of knottins in the KNOTTIN database, unrelated to LhKNOT, but representative of various taxa (cone snail, horseshoe crab, insect, plant, scorpion, and spider), showed the CPC clip motif. In all cases, the surface electrostatic profile of the CPC clip motif is positively charged, likely, to facilitate interactions with the negatively charged heparin (Figure 4). However, similar to the beetle, fungal, and wasp knottins described above, the CPC clip was not identified in knottins from the plant cyclotide and sponges (Appendix A).

### 2.5. In Silico Mutational Analysis of LhKNOT CPC Clip Reveals a Secondary CPC Clip

Models of three in silico LhKNOT mutants (R1A, S3A, R14A) were investigated to compare binding interactions of wild type and mutant proteins. In the case of the R14A mutant, heparin was shown to surprisingly associate with a secondary CPC clip involving the R1, S3, R30 residues instead of the R1, S3, R14 motif in the wild type (Appendix A). Double or triple mutants (R14A-R1A; R14A-S3A-R1A), however, did not show an interaction with any of the primary or secondary CPC clip motif residues (Appendix A). The in silico mutants with the R1A mutation (interrupting both the primary and secondary CPC clip motifs), however, did show heparin associating with R29 and R30, as they provide the only remaining cationic surface (Appendix A). Interestingly, the ligand prefers to associate with the CPC clip motif, both in the case of the primary and secondary clips, over this additional cationic surface when the motif is available, suggesting the specific interaction of the CPC clip may allow a more stable association over a non-specific electrostatic interaction.

### 2.6. A CPC Clip Is Also Present in the Antimicrobial Peptide Drosomycin

Unlike LhKNOT, the structure of Drosomycin contains a different fold with an alpha-helix and a twisted three-stranded beta-sheet, which is stabilized by three disulfide bridges [43]. This structural motif, termed a “cysteine stabilized αβ-motif”, is also found in other host-defense proteins such as the antibacterial insect defensin A, scorpion toxins, plant thionins, and antifungal plant defensins [43,44]. Given the known antimicrobial function of Drosomycin and its related “cysteine stabilized αβ-motif” [43,44], we examined the possibility of a common basis for antimicrobial activity and shared structure–function relationships with LhKNOT to better understand the potential significance of this motif in LhKNOT in the context of host defense. Despite a slightly different structural fold, Drosomycin overlays within an RMSD of 2.2Å of both LhKNOT and pB1 (Figure 5A). When the previous CPC clip analysis was extended to Drosomycin, the results revealed the presence of the CPC clip motif in Drosomycin as well. The CPC clip motif in Drosomycin (R6-S4-K38) falls within a 2Å deviation of the motif parameters (Figure 5B, Appendix A). Additionally, surface electrostatics of Drosomycin confirmed the putative CPC clip’s formation of a cationic binding pocket as also observed for LhKNOT and other knottins (Figure 5C, Appendix A). These results suggest that there may be similarity in the mode of action of Drosomycin and LhKNOT and that the biochemistry embedded in this structural motif may drive antimicrobial function in both peptides.

## 3. Discussion and Speculation

Biochemical functions of proteins are facilitated by their overall structures, which are reportedly 3–10 times more well-conserved than their primary structures [45]. Consequently, sequence alignments often fall short in investigating the functional potential of proteins such as knottins, where the varied primary structures may obscure shared structure–function relationships. In silico methods facilitate broad analysis of potential structure–function correlations for such interesting folds [46,47,48,49,50]. Our detailed analysis of the LhKNOT structure and its structural homologs supports the idea that this parasite protein has the hallmarks of a knottin peptide with functions in host defense: (a) the closest matches to LhKNOT are antimicrobial peptides; (b) LhKNOT’s Pfam signature describes it as cysteine knotted “antifungal peptide”; and (c) a CPC clip is present.

Analysis of a knottin wound-healing peptide bleogen pB1 from the leafy cactus *Pereskia bleo* initially revealed the presence of a CPC clip, a conserved structural signature of heparin-binding proteins [42]. It has immunity- and signaling-related functions in wound healing and cancer and is used as an anticoagulant [51]. GAGs are present on animal cell surfaces and have become an increasingly popular target of studies ranging from clinical to cosmetic [52]. The CPC clip motif of heparin-binding proteins is considered to be the minimum required motif for heparin-binding affinity and is defined by specific spatial relationships between three residues, two cationic (Arg/Lys) and one polar (Asn, Gln, Ser, Thr, Tyr). This CPC clip motif is defined by distances between both the α carbons of the involved residues as well as the distances between the centers of mass of the involved residues. Heparin-binding proteins are known to interact promiscuously with other negatively charged substrates, most notably the surface membranes of microbes [41]. As a result, heparin-binding proteins can exhibit antimicrobial activity to varying degrees [53]. While the precise mechanisms of this antimicrobial potential are not well defined, it has been hypothesized that the CPC clip may facilitate these protein–microbe interactions [41]. New potential roles of heparin binding are emerging, such as in the potential therapeutic interventions for amyloidogenesis [54], so understanding the structural determinants of the CPC clip in diverse proteins may be key to delineating the mechanism of the interaction and its disruption.

A clue to a possible antimicrobial mechanism for LhKNOT comes from our finding that its inhibitory cysteine knot and those of other knottins examined here have a CPC clip intrinsic to their folds (Appendix A). It has been noted that the structural and biophysical requirements for heparin-binding proteins’ ability to interact with heparin are strikingly similar to the structural features of many known AMPs [53]. Thus, the CPC clip motif may fit the criteria for both these requirements [41]. Our discovery of this CPC clip motif not only in LhKNOT but also in 75 other related or unrelated knottin proteins (Appendix A) suggests that this CPC clip motif may be a highly conserved feature of the knottin proteins. The coincident occurrence of the CPC clip motif within the cysteine knot structure in diverse proteins suggests that the CPC clip motif itself may underlie the antimicrobial function. However, because this structural motif is also identified in offensive knottins (Appendix A), it may also serve a more general role via this more generalized CPC clip interaction motif.

The discovery of a putative CPC clip motif in Drosomycin is intriguing, as it sheds light on the possible mechanism of action of this antifungal peptide and supports a possible antimicrobial function for the CPC clip motif itself. Previous studies have shown the importance of Drosomycin’s residues R6 and K38, which are part of its CPC clip motif (R6-S4-K38), for its antifungal potential; alterations of these residues reduce antifungal activity [55]. Our structural analysis of the wasp LhKNOT and its similarities with *Drosophila* Drosomycin offers possible insights into the *L. heterotoma*’s ability to broadly suppress Toll-NF-κB pathway in *D. melanogaster* and yet succeed on immune-compromised hosts. In the absence of a functional Toll immune pathway, and consequently without Drosomycin production, *Drosophila* larvae can often succumb to opportunistic pathogens [11,13,56]. LhKNOT’s putative antimicrobial activities might defend the host from opportunistic microbial pathogens in wasp-infected, immune-compromised host larvae and developing pupae. Alternatively, similar to Drosomycin, LhKNOT could induce hemocyte apoptosis [57]. *L. heterotoma* EVs enter hemocytes after infection [58], and their activities within EVs compromise encapsulation [19,21].

The CPC clip has been shown to associate promiscuously with GAGs [41], which are nevertheless ubiquitous in insects including *Drosophila* [59]. Heparin sulfate GAGs in flies serve as growth factor receptors and participate in creating and maintaining morphogenic gradients [60]. The CPC clip has also been shown to associate promiscuously with lipopolysaccharides (LPS) [41]. Bacterial LPS, being major surface components of Gram-negative bacteria, are extremely potent stimulators of the innate immune response in various eukaryotic species including insects [61]. GAGs in *Drosophila* also control the binding of α C protein, a virulence determinant of group B streptococcus [62]. Thus, it is possible that a CPC clip containing LhKNOT and Drosomycin (and possibly other fly AMPs, such as Defensin [63]) can modulate immune signaling (and/or binding to the pathogen’s virulence proteins), similarly affecting the bacterial pathogen/metazoan parasite outcome. Previous studies have shown that three homologous knottins (Alo-1, 2, 3) from the harlequin beetle *Acrocinus longimanus* are active against yeast, *Candida glabrata* [30]. Additionally, a pore-forming candidacidal peptide, *Psacotheasin*, attacks *Candida albicans* [64]. Lastly, a plant pathogenic rust fungus, *Melampsora larici-populina*, produces a knottin peptide as its own candidate effector [65].

Our finding that multiple putative knottin peptides may be secreted by adult parasitoid wasps of *Drosophila* was unexpected and intriguing. While their functions are not known, the possible existence of a knottin gene family suggests some redundancy in their functions that may be linked to the parasitic life history. The exploration of the knottin structural fold in unrelated and yet-to-be discovered knottins will prove to be valuable. Our work provides a short list of residues significant to the proposed physiological activity for knottin family protein investigation. Such information can be incorporated into rational drug discovery and design for infectious diseases and related therapies.

## 4. Methods

### 4.1. Sequence Analysis

Initial scans of the NCBI non-redundant protein sequence (nr) database (https://www.ncbi.nlm.nih.gov/protein; accessed on 4 April 2018), the Transcriptome Shotgun Assembly (TSA) sequences database (https://www.ncbi.nlm.nih.gov/genbank/tsa; accessed on 16 April 2018) [66], and the Protein Data Bank (PDB; https://www.rcsb.org; accessed on 5 July 2018) [67] with the full-length amino acid sequence of the *L. heterotoma* knottin peptide (GAJC01011813.1) using NCBI BLAST (BLASTp; https://blast.ncbi.nlm.nih.gov/Blast.cgi; accessed on 16 April 2018) [68] were used to retrieve related protein sequences in other organisms. Remote homologs were searched using Position-Specific Iterative BLAST (PSI-BLAST; https://blast.ncbi.nlm.nih.gov/Blast.cgi; accessed on 4 April 2022) [69], and Hidden Markov Models (HMMER; https://www.ebi.ac.uk/Tools/hmmer; accessed on 6 April 2022) [70,71]. BLAST searches were adjusted to word size 2, and scoring matrix BLOSUM45, while default parameters were used for HMMER.

To identify knottin homologs in select *Leptopilina* and *Ganaspis* spp. not accessible in the protein databases, targeted searches using translated BLASTs against genomic and transcriptomic datasets were employed. To identify additional *L. heterotoma* knottins, the *L. heterotoma* female abdominal transcriptome GAJC [72] and the whole-body transcriptomes from Ground GHUQ (GenBank Accession: GHUP00000000;TSA: *Leptopilina heterotoma* strain *Lh14*, transcriptome shotgun) and Space GHUP (GenBank Accession: GHUQ00000000;TSA: *Leptopilina heterotoma* strain *Lh14*, transcriptome shotgun) were searched using tblastn (e-value threshold set to 1, BLOSUM62, low complexity filter enabled, query coverage filter set to 20 percent) against LhKNOT. The following transcriptomes were similarly searched to identify *L. boulardi* knottins: *L. boulardi* female abdominal transcriptome GAJA [72] and the whole-body transcriptomes from Ground GITC (GenBank Accession: GITC00000000; TSA: *Leptopilina boulardi* strain *Lb17*, transcriptome shotgun), Space GISX (GenBank Accession: GISX00000000; TSA: *Leptopilina boulardi* strain *Lb17*, transcriptome shotgun), and GGGI00000000 (female abdomen/head) [73]. In addition, targeted searches for *Ganaspis* spp. homologs were carried out. For *G. hookeri*, the GAIW00000000 TSA: *Ganaspis* spp. G1 dataset [74] was searched (tblastn, e-value threshold set to 1, BLOSUM45, low complexity filter enabled, query coverage filter set to 20 percent) using the LhKNOT query. The Expasy tool (web.expasy.org/translate/; accessed on 9 December 2022) was used to translate these TSA sequences [75]. The resulting predicted peptide sequences were manually curated based on the presence of the cysteine motif. For *G. brasiliensis*, AUGUSTUS-based (http://bioinf.uni-greifswald.de/augustus; accessed on 9 April 2020) [76] gene predictions from the partially assembled genomic sequence (Gen-Bank:GCA_009823575.1) were made using the *Nasonia vitripenis* genome training module. The resulting predicted proteins were searched using blastp (e-value threshold set to 1, low complexity filter enabled, query coverage filter set to 20 percent) against LhKNOT. Alignments and visualization were performed using EBI’s Clustal Omega (https://www.ebi.ac.uk/Tools/msa/clustalo; accessed on 9 December 2022) [77] to confirm the presence of the knottin motif and for further curation of the sequences. The set of identified sequences were aligned to LhKNOT to create a multiple sequence alignment using T-COFFEE (https://www.ebi.ac.uk/Tools/msa/tcoffee; accessed on 9 December 2022) [78] and visualized using ESPript3 (https://espript.ibcp.fr/ESPript/ESPript/index.php; accessed on 9 December 2022) [79] to identify any conserved residues and/or motif(s) in the aligned sequences.

### 4.2. LhKNOT Primary Structure Analysis

The LhKNOT sequence was also analyzed by the domain/motif detection database Conserved Domain Database (CDD; https://www.ncbi.nlm.nih.gov/Structure/cdd/cdd.shtml; accessed on 6 April 2018) and Pfam (pfam.xfam.org; accessed on 6 April 2018) to identify any domains or sequence signatures [80,81]. The sequence was further analyzed using the signal sequence prediction software SignalP-5.0 (https://services.healthtech.dtu.dk/service.php?SignalP-5.0; accessed on 6 April 2018) [82] and Phobius (https://www.ebi.ac.uk/Tools/pfa/phobius; accessed on 6 April 2018) [83] to identify the location of the signal sequence and allow for a more accurate characterization of the peptide. The mature peptide of 36 amino acids (with the predicted signal sequence, residues 1–24, removed) was used for further analysis. The secondary structure of the peptide was predicted using the meta-server Sympred (https://www.ibi.vu.nl/programs/sympredwww; accessed on 6 April 2018) [84] and analyzed to confirm its correspondence to the expected secondary structure for a knottin fold: an initial alpha-helical segment (predicted signal sequence) followed by three anti-parallel beta strands.

### 4.3. Tertiary Structure Prediction and Evaluation of LhKNOT and Its Homologs

The programs I-Tasser (https://zhanggroup.org/I-TASSER; accessed on 16 April 2018) [85], Modeller (https://salilab.org/modeller; accessed on 16 April 2018) [86], and HHPred (https://toolkit.tuebingen.mpg.de/tools/hhpred; accessed on 6 April 2018) [37] were used to model the tertiary structure of LhKNOT. Three structural homologs with the knottin-like fold Alo-3 (PDB: 1Q3J) [30], omega-agatoxin-IVA (PDB: 1IVA) [28], and PAFP-S (PDB: 1DKC) [25] were identified as top candidates and used as templates in model building programs. The predicted model for LhKNOT was evaluated by Verify3D (https://www.doe-mbi.ucla.edu/verify3d; accessed on 18 April 2018) [39], VoroMQA (https://bioinformatics.lt/wtsam/voromqa; accessed on 18 April 2018) [40], and Prosa-web (https://prosa.services.came.sbg.ac.at/prosa.php; accessed on 18 April 2018) [38]. Evaluating small peptides with standard evaluation tools can have limitations. Thus, scores from solved tertiary structures of NMR and crystallography-derived peptides of similar size and shape (PDB: 1Q3J, 1IVA, and 1DKC) were used to interpret the evaluation scores of our LhKNOT models. The top-ranked model was created using HHPred [37], a modeling program that uses the underlying model-building program, Modeller [86] using a restraint-based comparative modeling approach. In addition, the KNOTTIN database’s KNOTER3D tool (https://www.dsimb.inserm.fr/KNOTTIN/knoter3d.php; accessed on 15 May 2020) [36] was used to evaluate the model’s tertiary structure, confirming LhKNOT’s adoption of the Knottin fold. All the identified homologs with no known structures were modeled using the top template identified in HHPred [37].

### 4.4. CPC Clip Motif Identification

Superposition of LhKNOT’s structural model with the known structures of a CPC clip containing knottin peptide from the cactus *Pereskia bleo* (PDB: 5XBD) [26] using the programs Superpose (http://superpose.wishartlab.com; accessed on 12 September 2018) [87] and MultiProt (http://bioinfo3d.cs.tau.ac.il/MultiProt; accessed on 12 September 2018) [88] suggested the presence of a heparin-binding CPC clip motif. To confirm the presence of the CPC clip in LhKNOT, docking analysis using the program, ClusPro 2.0 (which offers parameters specific to heparin as ligand; https://cluspro.org; accessed on 2 March 2019) [89], and subsequent analysis of predicted docking scenarios in the visualization program, PyMOL (https://pymol.org; accessed on 10 March 2019) [90] was performed. All potential docking scenarios were analyzed for identifying residues forming the putative CPC clip motif using the following steps. All polar interactions between LhKNOT and heparin were visualized in PyMOL and lists of participating residues were compiled. Predictions that contained the necessary Cation–Polar–Cation interactions were then further scrutinized for adherence to the CPC clip motif parameter by measuring distances between alpha carbons of participating residues as well as distances between center of mass of participating residues. PyMOL’s “Measurements” Wizard tool was used to measure distances between α-carbons of the participating residues. To evaluate the distances between centers of mass, a user-generated script available in the pymolwiki website (Henschel, https://pymolwiki.org/index.php/Center_of_mass; accessed on 15 March 2019) was used to generate pseudo-atoms at the calculated sidechain center of mass. PyMOL’s APBS Electrostatics Plugin [91] was used to render the surface electrostatics for the modeled LhKNOT. Similar steps were carried out to identify the CPC clip motif in each of the remaining knottin proteins in Appendix A.

### 4.5. Association of LhKNOT with Other GAGs

To assess whether the CPC clip would facilitate binding with additional GAGs, further docking was performed with hyaluronic acid (PDB: 1HYA) and keratan sulfate (PDB: 1KES). In the case of hyaluronic acid, extraneous atoms (O, Na) were removed from the space surrounding the molecule. Keratan sulfate’s model was used as is. Docking was performed using AutoDock (https://autodock.scripps.edu; accessed on 7 April 2020) [92], and results were visualized in PyMOL. Results were then input into PDBSum (http://www.ebi.ac.uk/thornton-srv/databases/pdbsum/Generate.html; accessed on 10 April 2020) [93] to generate Ligplots to confirm GAG-CPC clip binding.

### 4.6. Protein-Heparin Docking Analysis for Mutated CPC Clip in LhKNOT

To investigate the importance of the CPC clip in heparin binding, mutated models of LhKNOT were generated using PyMOL’s mutagenesis tool, substituting the CPC clip residues with alanine. Models were created piecewise, first substituting just R14, then R1 and R14, and finally R14, R1, and S3. Heparin docking of all models was performed using ClusPro, and analysis was performed as described previously.

### 4.7. Protein-Heparin Docking Analysis in Structural Homologs and Drosomycin

To further evaluate the presence of the CPC clip motif in the knottin peptide family, the previous methods were further expanded to (a) structural homologs of LhKNOT (PDB: 5XBD, 1Q3J, 1IVA, and 1DKC) [25,26,28,30]; (b) *Drosophila* Drosomycin (PDB: 1MYN) [43], a peptide with known antifungal activity; (c) modeled homologous knottin proteins, and (d) representative knottins from different categories in the KNOTTIN database. Results were considered to adhere to the motif if measurements fell within 3Å of the CPC clip parameters, to allow for some flexibility of protein loop conformations.

## Figures and Tables

**Figure 1 pathogens-12-00143-f001:**
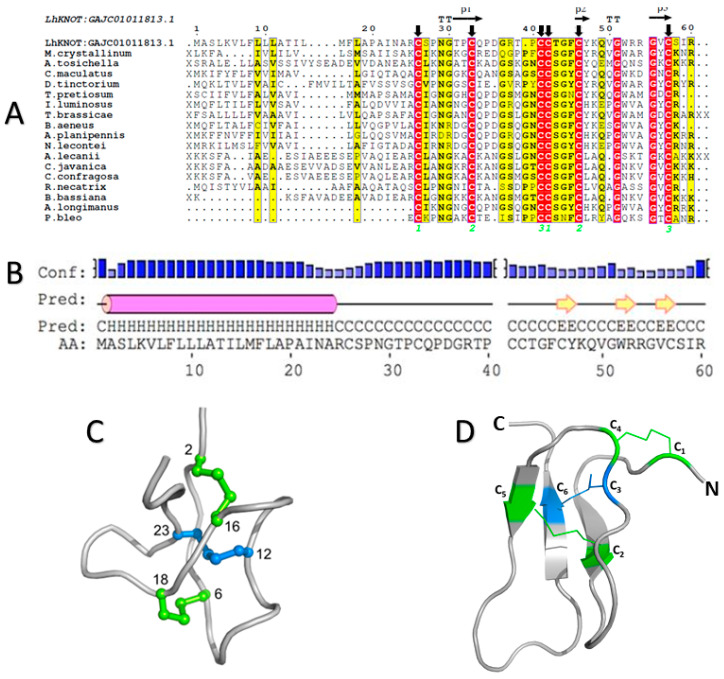
(**A**) Multiple sequence alignment of LhKNOT (top) and homologs in the knottin family (top-to-bottom): *Mesembryanthemum crystallinum* (plant; antimicrobial; NCBI ID: AAC19399), *Aceria tosichella* (mite; antimicrobial; NCBI ID:MDE48292.1), *Callosobruchus maculatus* (beetle; unknown; NCBI ID:VEN35376.1), *Dinothrombium tinctorium* (mite; unknown; NCBI ID:RWS16713.1), *Trichogramma pretiosum* (Alo-2 like; parasitoid wasp; antimicrobial; NCBI ID: XP_014233229.1), *Ignelater luminosus* (beetle; unknown; NCBI ID: KAF2884324.1), *Trichogramma brassicae* (parasitoid wasp; unknown; NCBI ID: CAB0031014.1), *Brassicogethes aeneus* (beetle; unknown; NCBI ID: CAH0563829.1), *Agrilus planipennis* (ash borer; antimicrobial; NCBI ID: XP_018321119.1), *Neodiprion lecontei* (sawfly; unknown; NCBI ID: XP_015517007.2), *Akanthomyces lecanii* (fungus; antifungal; Uniprot ID: A0A168C5L6), *Cordyceps javanica* (fungus; antifungal; Uniprot ID: A0A545VVU1), *Cordyceps confragosa* (fungus; unknown; Uniprot ID: A0A179IJT7), *Rosellinia necatrix* (fungus; unknown; Uniprot ID: A0A1S8A723), *Beauveria bassiana* (fungus; unknown; Uniprot ID: A0A0A2VUF6), *Acrocinus longimanus* (Alo-3; beetle; antifungal; PDB ID: 1Q3J), and *Pereskia bleo* (pB1; cactus; PDB ID: 5XBD). Identical residues are highlighted in red, while chemically similar residues are highlighted in yellow. Black arrows denote the conserved cysteine residues that are responsible for the knottin structural fold, and the disulfide bridge connectivity of the three disulfide bridges is marked with neon green numbers. X and XX denote the variable number of residues at the N- and C-terminus, respectively. Secondary structure elements of the modeled LhKNOT are shown at the top. β and T represent β-strand, and β-turn, respectively. (**B**) Secondary structure prediction for LhKNOT, showing a long helical segment (pink cylinder; signal peptide) followed by three beta strands (yellow arrows). This secondary structure follows the canonical knottin signature [36]. (**C**) A prototype of the knottin structural fold from the KNOTTIN database [36] (permission to reproduce this image given by Dr. Jean-Christophe Gelly, INSERM, October 2019). The two disulfide bridges (green) form a macro-cycle, while a third (blue) pulls through the center giving the protein its namesake knotted topology. (**D**) Predicted tertiary structure of LhKNOT, showing the knotted topology and conserved cysteine motif, cysteine residues form disulfide bridges in the conserved pattern seen throughout the knottin family (C_1_-C_4_ and C_2_-C_5_ form the macro-cycle, C_3_-C_6_ pulls through the center).

**Figure 2 pathogens-12-00143-f002:**
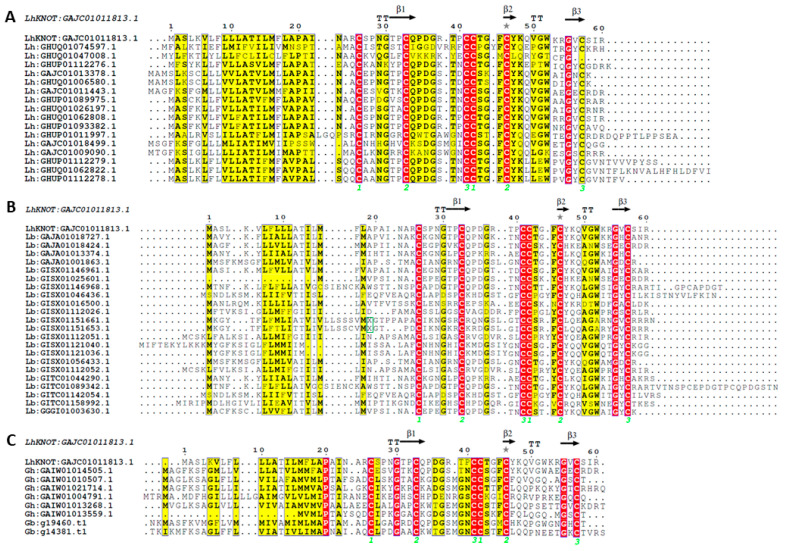
(**A**) Multiple sequence alignment of LhKNOT (top) and homologs in *Lh14*. (**B**) Multiple sequence alignment of LhKNOT (top) and homologs in *L. boulardi*. The X marked in the green box corresponds to the following insertions in two *Lb17* sequences: Lb:GISX01151661.1 X = SQAPPPTPEPFHPPGTPP; Lb:GISX01151653.1 X = SQLTSPRSTFPPTGSTIPSIITT. (**C**) Multiple sequence alignment of LhKNOT (top) and homologs in *Ganaspis* spp. Identical residues are highlighted in red, while chemically similar residues are highlighted in yellow. Black arrows denote the conserved cysteine residues that are responsible for the knottin structural fold, and the disulfide bridge connectivity of the three disulfide bridges is marked with neon green numbers. Secondary structure elements of the modeled LhKNOT are shown at the top. β and T represent β-strand, and β-turn, respectively. Grey stars added on the top of blocks of sequences, above residues represent residues modeled with alternate conformations.

**Figure 3 pathogens-12-00143-f003:**
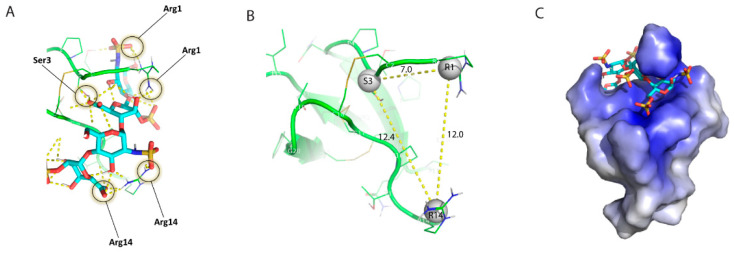
(**A**) Visualization of the CPC clip motif in LhKNOT. Hydrogen bonds between heparin (teal) and participating residues (Arg1, Ser3, Arg14) of LhKNOT (green) are indicated by dashed yellow lines. (**B**) Distances measured in PyMOL between the centers of mass of participating residues (grey pseudo-atoms) in the LhKNOT–heparin interaction. Distances are marked on the yellow dashed lines and fall within the expected value ranges of the CPC clip motif. (**C**) Surface electrostatics of LhKNOT, in complex with heparin. The positively charged surface formed by the cationic residues involved in the CPC clip allows for electrostatic interactions with the overall negatively charged heparin molecule. The surface electrostatic potentials are color-graded from −4 kT/e (red) to +4 kT/e (blue).

**Figure 4 pathogens-12-00143-f004:**
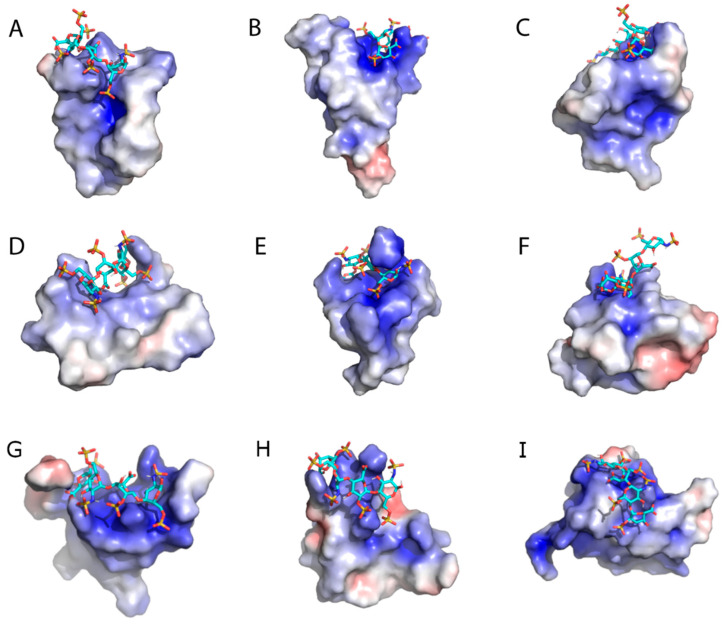
Heparin-docking results and surface electrostatics of (**A**) antifungal peptide of *Phytolacca americana* (PFAP-s; plant; PDB: 1DKC), (**B**) calcium channel selective omega-agatoxin of *Agelenopsis aperta* (Omega-agatoxin IV-A; spider; PDB: 1IVA), (**C**) antifungal peptide of *Acrocinus longimanus* (Alo-3; beetle; PDB: 1Q3J), (**D**) wound-healing peptide of *Pereskia bleo* (pB1; cactus; PDB: 5XBD), (**E**) LhKNOT, (**F**) antifungal peptide “Drosomycin” of *Drosophila* spp. (Drosomycin; fly; PDB: 1MYN), (**G**) Conotoxin GS (Conotoxin GS; cone snails; PDB:1AG7), (**H**) antimicrobial peptide Tachystatin A isolated from Horseshoe crabs (Tachystatin A; horseshoe crabs; PDB:1CIX) and (**I**) Chlorotoxin, a small scorpion toxin of *Leiurus quinquestriatus* (Chlorotoxin; scorpion; PDB:1CHL) showing the cationic CPC clip motif. Figures oriented to best display the CPC clip structure. The surface electrostatic potentials are color-graded from −4 kT/e (red) to +4 kT/e (blue).

**Figure 5 pathogens-12-00143-f005:**
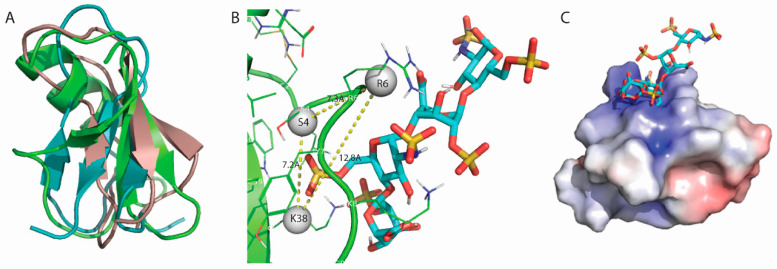
(**A**) Superposition of LhKNOT (blue), wound-healing peptide of *Pereskia bleo* (tan; pB1;cactus; PDB: 5XBD) and Drosomycin (green; Drosomycin; fly; PDB: 1MYN). (**B**) Distances measured in PyMOL between the centers of mass of participating residues (grey pseudo-atoms) in the Drosomycin–heparin interaction. Distances fall within 2Å of the value ranges of the CPC clip motif. (**C**) Surface electrostatics of Drosomycin (fly; PDB: 1MYN), showing the positively charged binding surface formed by the CPC clip where heparin is docked. The surface electrostatic potentials are color-graded from −4 kT/e (red) to +4 kT/e (blue).

## Data Availability

The data presented in this study are available in the article and Appendix A.

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
