# Peer review of "In Silico Analysis of a Drosophila Parasitoid Venom Peptide Reveals Prevalence of the Cation–Polar–Cation Clip Motif in Knottin Proteins"

_pathogens, 2023, doi:10.3390/pathogens12010143_

Round 1

Reviewer 1 Report

Dear Authors,

The manuscript is interesting and shows the bioinformatics results of LhKNOT’s-residue core knottin and their possible defensive function for the parasitoid LhKNOT (physiology and the host–pathogen recognition process).

 -        The work is well designed and written; a minor grammatical revision is required throughout the text; for example, some articles, commas, adverbs, etc., are missing.

-        Throughout the text, check the conjugation of the verbs.

-        The paper needs to be checked carefully for typos and grammatical errors.

-        The scientific names should be written in italics.

-        Include the web address of all servers and databases.

-        Keep the same font style and size throughout the manuscript.

-        In table 1, the database name of accession numbers is missing.

-        In table 4, the column of species name, protein, and function, I suggest dividing it into 3 columns. It is also necessary to include the bibliographical references of the functions.

-        In table 4, the database name of accession numbers is missing.

-        At the reference list, some names of species are not italics.

L31-L33. I suggest changing the phrase to The CPC clip is likely a highly conserved structural motif found in many diverse proteins with reported heparin binding capacity, including amyloid proteins.

L268. I suggest eliminate the word participating may be unnecessary in this sentence. Consider remove it.

L3434. I suggest write: knottin defensive instead defensive knottin.

L354. The phrase it is quite possible that, may be unnecessary in this sentence. Consider remove it.

Some significant points should be addressed before the manuscript can be considered for publication.

Reviewer 2 Report

The manuscript entitled „In silico Analysis of a Drosophila Parasitoid Venom Peptide Reveals Prevalence of the Cation-Polar-Cation Motif in Knottin Proteins” by Arguelles et al. provides results from an in-silico approach to identify motifs in the LhKNOT peptide with homologies to antimicrobial peptides. The study is interesting and important for the identification and characterization of such peptides.

However, the manuscript is partially difficult to read and I would propose a re-organization of the manuscript.

Introduction:

This section provides a comprehensive overview about antimicrobial defense peptides in insects. However, the overall aim of the study should be more clearly defined in the introduction. Moreover, parts of the introduction are already describing results and would better fit to the results or discussion section. I would recommend to provide only a brief summary of the findings at the end of the introduction and leave the rest for the discussion.

Results:

I would recommend to re-organize the results section:

·        List of alignments of LhKNOT in Table 1 would better fit to the supplementary information. Rather, alignments with selected sequences as shown in Fig. 1A would be more informative, in particular also with those which show lower homologies.

·        Line 136 should read: “…showed similar results with sequences from common ice plant…”

·        Line 145: please add protein names, otherwise it is a bit confusing, such as

“…for the knottin-type antifungal peptide Alo-3 from the insect Acrocinus longimanus and the second from the disulfide-constrained wound healing peptide pB1 from Pereskia bleo

·        Tables 2 and 3 could be also shown in the Supplementary information. A pure listing of the sequence ID´s is not very informative. Again, it would be more interesting to see examples of translated protein sequences

·        Figure 1: protein names should be added in the legend (also in other figures) Phytolacca americana (PAFP-S; plant; antifungal; PDB: 1DKC), Acronicus longimanus 177 (Alo-3; beetle; antifungal; PDB: 1Q3J) and Pereskia bleo (pB1; cactus; PDB: 5XBD).

·        Table 4 is much too long and would also fit better the supplementary information. The main emphasis should be on relevant examples as shown in the figures.

·        Paragraph line 292: “In silico mutational analysis of LhKNOT CPC clip reveals a secondary CPC Clip” does not refer to any figure, where is that shown? If there is no data to be shown, this paragraph should be added to the previous section and not be in a separate section with an own header but lacking data.

Round 2

Reviewer 2 Report

The manuscript entitled „In silico Analysis of a Drosophila Parasitoid Venom Peptide Reveals Prevalence of the Cation-Polar-Cation Clip Motif in Knottin Proteins” by Arguelles et al. has been improved according to the suggestions and is now acceptable for publication.

There are only a few additional comments:

-        The manuscript has two running titles

-        In Figure 2 A, B, C, the text has different sizes – maybe these can be adjusted? In Fig. 2B longer sequencing stretches from two examples (after residue 20 and before first cysteine could be also replaced by an X?)

-        Line 227: Should be Figure 3A instead of Figure 2A
